

# Dominant forest tree species are potentially vulnerable to climate change over large portions of their range even at high latitudes

Catherine Périé[1] and Sylvie de Blois[2,3]

[1] Direction de la Recherche Forestière, Ministère des Forêts, de la Faune et des Parcs, Québec, Canada
[2] Department of Plant Science, Macdonald Campus, McGill University, Ste-Anne-de-Bellevue, Quebec, Canada
[3] McGill School of Environment, McGill University, Montreal, Quebec, Canada

## ABSTRACT

Projecting suitable conditions for a species as a function of future climate provides a reasonable, although admittedly imperfect, spatially explicit estimate of species vulnerability associated with climate change. Projections emphasizing range shifts at continental scale, however, can mask contrasting patterns at local or regional scale where management and policy decisions are made. Moreover, models usually show potential for areas to become climatically unsuitable, remain suitable, or become suitable for a particular species with climate change, but each of these outcomes raises markedly different ecological and management issues. Managing forest decline at sites where climatic stress is projected to increase is likely to be the most immediate challenge resulting from climate change. Here we assess habitat suitability with climate change for five dominant tree species of eastern North American forests, focusing on areas of greatest vulnerability (loss of suitability in the baseline range) in Quebec (Canada) rather than opportunities (increase in suitability). Results show that these species are at risk of maladaptation over a remarkably large proportion of their baseline range. Depending on species, 5–21% of currently climatically suitable habitats are projected to be at risk of becoming unsuitable. This suggests that species that have traditionally defined whole regional vegetation assemblages could become less adapted to these regions, with significant impact on ecosystems and forest economy. In spite of their well-recognised limitations and the uncertainty that remains, regionally-explicit risk assessment approaches remain one of the best options to convey that message and the need for climate policies and forest management adaptation strategies.

## INTRODUCTION

During the last century, forest conservation policies and management practices worldwide have been developed assuming a relatively stable climate regime. Indeed, apart from occasional extreme events, climate was largely considered as a stable dimension, over

Corresponding author
Catherine Périé,
catherine.perie@mffp.gouv.qc.ca

decades or centuries, of a species' niche or habitat. Although tree species distribution ranges have expanded or shrunk in response to climate, detectable shifts largely occurred at time scales comparable to those of climate change in the Quaternary, that is, within centuries or millennia for long-lived trees (*Davis, Shaw & R*, *2005*). In the coming decades, however, boreal forests are predicted to face multiple stresses under a rapidly warming climate, including increased frequency of forest fires and insect and disease outbreaks (*Gauthier et al.*, *2015*). Global mean temperatures are projected to increase at rates unprecedented in human history (*Diffenbaugh & Field*, *2013*). By the mid-21st century, many areas of the globe will be under a new climate regime, in which the coolest warm-season months of the 21st century are predicted to be hotter than the hottest warm-season months of the late 20th century (*Diffenbaugh & Scherer*, *2011*), while considerable regional and interannual variability is expected. Impacts could be profound on forest species distributions, community structure, and ecosystem functions, as well as on all economic activities and services that depend on forests (*Hanewinkel et al.*, *2012*; *Price et al.*, *2013*).

Projecting suitable conditions for a species as a function of future climate provides a reasonable, although admittedly imperfect, spatially explicit estimate of tree vulnerability associated with climate change in this century (*Araújo & Peterson*, *2012*; *Elith & Leathwick*, *2009*; *Franklin*, *2013*). Species distribution or habitat suitability models have projected dramatic range shifts at continental scales for hundreds or thousands of species at a time, greatly helping raise concerns about biodiversity and climate change (*Iverson et al.*, *2008*; *Ray, Morison & Broadmeadow*, *2010*; *Thuiller et al.*, *2008*; *Xiao-Ying, Chun-Yu & Qing-Yu*, *2013*). Projections will usually show potential for areas to become climatically unsuitable, remain suitable, or become suitable for a particular species with climate change compared to baseline climatic conditions. Each of these outcomes, however, raises markedly different ecological and management issues. For instance, the potential for habitat gain under warmer climatic conditions exists but natural tree range expansion or tree migration is unlikely to proceed at rates sufficient to keep up with climate change in this century (*Boisvert-Marsh, Périé & De Blois*, *2014*; *Renwick & Rocca*, *2015*; *Savage & Vellend*, *2015*), whereas the introduction of species outside their natural range is questioned (*Aubin et al.*, *2011*). On the other hand, if climatic conditions are projected to become unsuitable for a species, many areas are likely to retain maladapted trees given trees' long lifespan; this could affect forest productivity and species turnover at a site. Species decline will have immediate consequences on local community processes, forest management practices, and related economic activities. Unless forests change mostly through catastrophic events, it is likely that managing forest decline at sites where climatic stress is becoming increasingly important will be the most immediate challenge of climate change. Finally, projections at continental scale that emphasize major range shifts may mask contrasting patterns at local or regional scale, while forest managers, conservationists, or policymakers need to understand site-specific impacts to inform adaptation strategies, forest policies, or monitoring efforts. Monitoring sites at risk, in particular, is increasingly important to determine whether recent climate change is already affecting population dynamics (*Girardin et al.*, *2014*; *Worrall et al.*, *2013*) or species distributions (*Boisvert-Marsh, Périé & De Blois*, *2014*; *Woodall et al.*, *2009*), or whether species can indeed persist under novel climatic conditions.

Here, we take advantage of available information on tree species distributions from forest survey programs in Quebec (Canada) and the eastern United States to assess potential decline in habitat suitability associated with climate change for five dominant tree species of deciduous and coniferous forests. Given the ecological and economic importance of these species, a change in their distribution and dynamics could make entire ecosystems, ecoregions, and economies vulnerable. Consequently, we focus on areas where climate is predicted to become unsuitable or less suitable for these species as opposed to habitat gain or range shift to emphasize vulnerabilities rather than opportunities. These species are, in order of decreasing merchantable volume in Quebec forests: *Picea mariana* (Mill.) Britton, Sterns & Poggenb. (black spruce), *Abies balsamea* (L.) Mill. (balsam fir), *Betula papyrifera* Marshall (white birch, synonym of paper birch), *Acer saccharum* Marsh. (sugar maple) and *Betula alleghaniensis* Britton (yellow birch). We base our assessment on a rigorous modelling approach using data spanning two jurisdictions (United States and Canada), but focus our interpretation at the scale of ecologically and economically significant bioclimatic domains which are defined by the target species in Quebec forests. We assume that (1) even though other factors can limit tree distribution (*Beauregard & De Blois*, *2014*; *Lafleur et al.*, *2010*), climate remains a significant determinant of a species' fundamental niche (*Araújo & Peterson*, *2012*), given its major role in determining species presence and genetic variation across landscapes (*Jansen et al.*, *2007*; *Woodward & Williams*, *1987*); (2) climate models coupled with greenhouse gas emission scenarios provide a reasonable estimate of climatic conditions in this century; (3) assessing potential decline in habitat suitability for a species provides an estimate of the risk of climate-related stress for that species; and (4) stakeholders need spatially explicit projections at a scale relevant to decision making, since trees regenerating today will cope with climate conditions that may drastically change during their lifespan. This is especially the case in boreal forests where most tree species grow slowly (*Ministère des Ressources Naturelles*, *2013*). We discuss the significance of these projections for species conservation and management scenarios, recognising the effect of uncertainty on adaptation strategies.

## MATERIALS & METHODS

### Study area

We focused our study on forests of Quebec (Canada), which account for 20% of the total Canadian forests and 2% of the world's forests. Dense forests cover an area of 761,100 km$^2$, (equivalent in size to the territories of Norway and Sweden combined— https://www.mern.gouv.qc.ca/english/international/forests.jsp), of which 70% is considered productive (commercial forest managed under the *Sustainable Forest Development Act*). Quebec forests are largely under public management (91.6% of forest land) with the responsible ministry allocating harvesting rights. The productive forest territory (45°N–53°N) mainly comprises the northern temperate and boreal vegetation zones (Fig. 1), which reflect Quebec's major climatic gradient. The zones are further divided, on the basis of edaphic and climatic conditions, into characteristic plant communities of ecological and economic importance or bioclimatic domains. The northern temperate zone includes, from south to

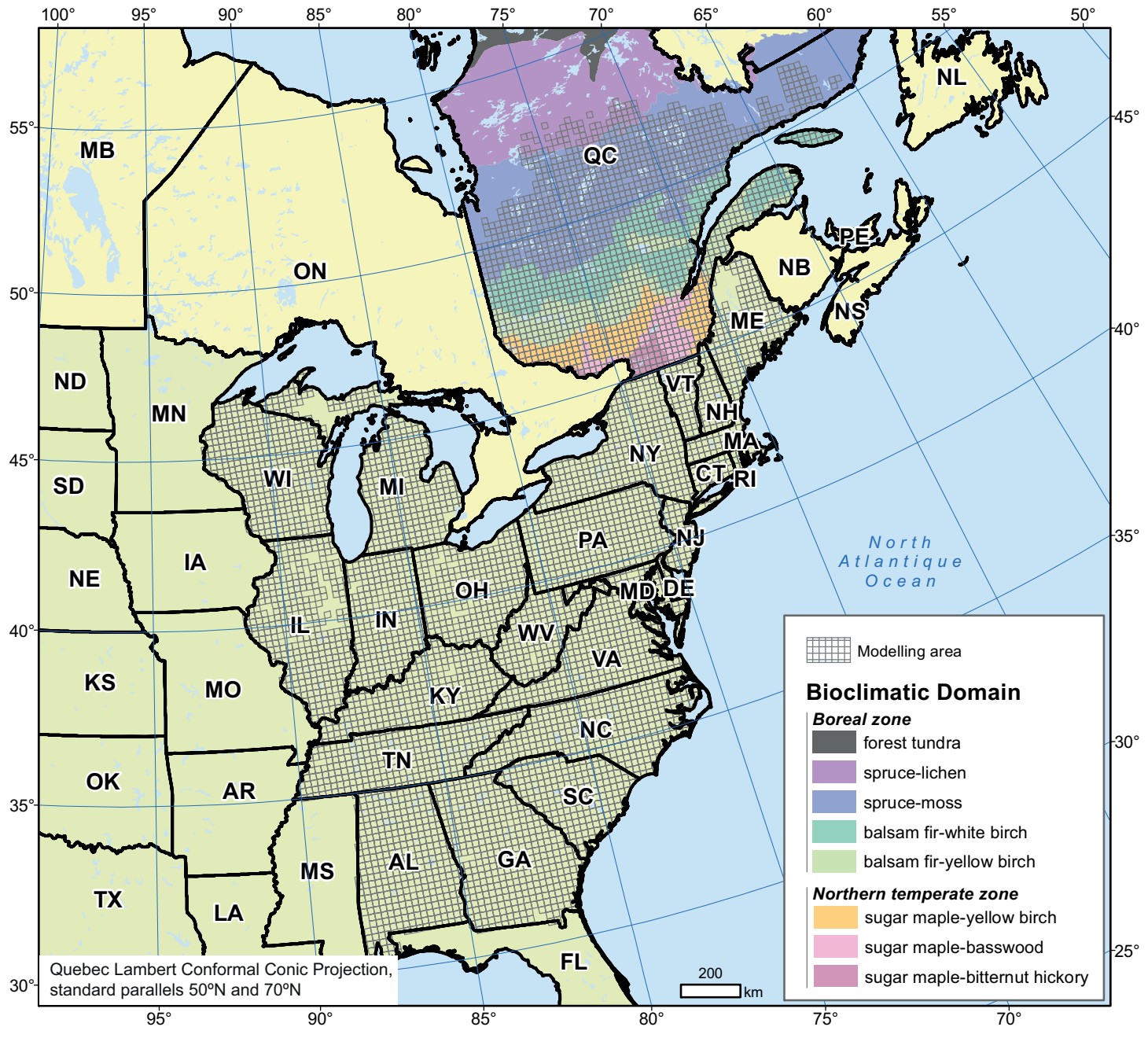

**Figure 1** Modelling area and spatial distribution of bioclimatic domains in Quebec (Canada).

north: the sugar maple–bitternut hickory domain (14,500 km$^2$), the sugar maple–basswood domain (31,000 km$^2$), the sugar maple–yellow birch domain (65,600 km$^2$)—all three being grouped in this study as the sugar maple domains—and the balsam fir–yellow birch domain (98,600 km$^2$). The boreal zone includes the balsam fir–white birch domain (139,000 km$^2$), the very large spruce–moss domain (412,400 km$^2$), and the spruce–lichen domain (299,900 km$^2$) which extends to 55°N.
We constructed habitat suitability models for each species using a modelling area largely exceeding that of the province taking into account the expected shift north of climate envelope according to various climate simulations (*Logan et al.*, *2011*) as well as available data on current species distribution and climate and edaphic conditions. The modelling area (~2,500,000 km$^2$) ranges from 30°N to 53°N in latitude and from 93°W to 60°W in longitude (Fig. 1). Annual mean temperature increases gradually from –5 °C in the north to +20 °C in the south, whereas annual total precipitation ranges from 670 to 2,000 mm, with less of a spatial gradient. Elevation ranges from sea level to 1,250 m.

We based the geographic grid we used for modelling on that of the *Climate Change Atlas for 134 Forest Tree Species of the Eastern United States* (*Iverson et al.*, *2008*; *Landscape Change Research Group*, *2014*). This grid was extended into Quebec to allow the merging of data sets from both jurisdictions. The mapped area is composed of 6,418 cells (20 × 20-km or 400-km$^2$ each, Fig. 1), each considered as a sampling unit and containing information on tree species occurrence, climate, elevation and edaphic characteristics.

Ideally, the modelling area should include the vast majority of the range of climatic conditions experienced by a species (*Barbet-Massin, Thuiller & Jiguet*, *2010*), including the range of projected climatic conditions according to scenarios of climate change in the study area. Covering the full distribution range of a species is not always possible and so, in preliminary analyses, we verified gaps in climate coverage for each species by comparing the modelling area with Little's range, which is assumed to cover an entire species range in North America (*Little Jr*, *1971*). There were minimal or no gap in temperature coverage for sugar maple (Fig. S1A), yellow birch (Fig. S2A), and balsam fir (Fig. S3A ; coverage of 100%, 100%, and 97% of the temperature range respectively). Temperature coverage was 70% for white birch (Fig. S4A) and 61% for black spruce (Fig. S5A), but the gaps were for colder temperatures that are not characteristic of the projected climate trends in the study area (+1.9 °C to +8 °C; Ouranos 2015). Precipitation coverage was 98% for both sugar maple (Fig. S1B) and yellow birch (Fig. S2B), and 70% for balsam fir (Fig. S3B). The gaps for white birch (35%; Fig. S4B) and black spruce (31%; Fig. S5B) were towards drier annual climates that are also not characteristic of the projected climate trends in the study area (precipitations +3% to +26%; *Ouranos*, *2015*).

## Occurrence data and target species

For the eastern United States, our main source of information was the Climate Change Atlas database (*Landscape Change Research Group*, *2014*). For the Canadian part of the modelling area, we obtained data on tree species occurrence from more than 95,000 forest plots sampled across the province and inventoried from 1985 to 1998 by the ministère des Forêts, de la Faune et des Parcs (Quebec's Ministry of Forests, Wildlife and Parks). The presence (or absence) of each forest tree species was recorded in each of the 6,418 cells of the modelling area.

The five target species (sugar maple, yellow birch, white birch, balsam fir, and black spruce) are common and widespread in the study area, and define the major bioclimatic domains described previously. The ecological and economic importance of these species cannot be overemphasized for the province: together, they represent 72% of the total

**Table 1  Predictor variables used in tree habitat suitability models.**

| Climatic | Topographic | Edaphic | |
| --- | --- | --- | --- |
| | | **Surface deposit** | **Drainage** |
| Annual mean temperature (°C) | Average elevation (m) | Eolian | Humid water regime |
| Annual total precipitation (mm) | | Fluvio-glacial | Mesic water regime |
| Ratio of summer precipitation over annual total precipitation | | Glacial | Xeric water regime |
| | | Littoral, marine or lacustre | |
| | | Organic | |
| | | Rocky substrate | |
| | | Slope or altered | |

volume of merchantable trees (Fig. S6), and many local economies are tightly linked to their fate. Their average longevity ranges from 150 years (balsam fir) to more than 300 years (sugar maple and yellow birch) (*Ministère des Ressources Naturelles, 2013*). Projections to the end of this century are thus well within their lifespan.

## Environmental data

We used 14 predictor variables for modelling (Table 1), including 3 climate, 1 elevation, 7 soil-class, and 3 soil property variables. The three climate variables were selected among a set of climate variables through cluster analysis (see below). Collinearity diagnostics measuring the relationship between the selected climate variables and other environmental variables were done using several methods (pairwise scatterplots, correlation coefficients, condition index, variance inflation factor). Collinearity was not found to be an issue at the scale of this study and for these sets of variables.

### Climatic data

Normalized (1961–1990) monthly surfaces of total precipitation and average, maximum, and minimum temperatures were downloaded from the USDA Forest Service Rocky Mountain station website (http://forest.moscowfsl.wsu.edu), as were other derived climatic variables  (see *Rehfeldt et al. (2006)* for more details). Data were obtained at a spatial resolution of 0.0083 decimal degrees (≈1 km) and averaged for each 20-km × 20-km grid cell of the modelling area. Climate variables tend to be highly correlated, so we used the VARCLUS procedure in SAS 9.2 (*SAS Institute Inc, 2008*) to find groups of variables that were as correlated as possible among themselves and as uncorrelated as possible with variables in other clusters. This analysis led to three clusters; in each cluster, we then selected one climate variable influencing plant survival and growth. They were mean annual temperature (TEM), mean annual precipitation (PRE) and useful precipitation (i.e., the ratio of the sum of June, July, August and September monthly precipitation to total annual precipitation; PRATIO).

Ouranos (http://www.ouranos.ca/en/), a consortium on regional climatology and adaptation to climate change, provided different climate simulations using output from 12 general and one regional coupled atmosphere–ocean general circulation models.

Each of these was coupled with 1, 2 or 3 projected greenhouse gas emissions scenarios (scenarios B1, A1B and/or A2, based on the Special Report on Emissions Scenarios, or SRES; http://www.ipcc.ch/ipccreports/sres/emission/index.php?idp=0). This generated a total of 70 climate simulations, which are a subset of the 86 climate simulations (*Logan et al.*, *2011*) made available from phase 3 of the Coupled Model Intercomparison Project (*Meehl et al.*, *2007*).

For each climate simulation, future (2071–2100) TEM, PRE and PRATIO values were obtained using the ''change field'' method (*IPCC*, *2001*). Monthly mean differences between the baseline period model run (1961–1990) and the future climate model run (2071–2100) were calculated and then combined with baseline values of the observed monthly climate data set. However, due to the relatively coarse spatial resolution of the climate simulations (45 km per cell side for the regional coupled atmosphere–ocean model, and ∼250-km per cell side for the general coupled atmosphere–ocean circulation models), monthly delta values for the centroids of each 20-km × 20-km grid cell (6,418 in all) were interpolated using a linear triangle-based interpolation method (*De Berg et al.*, *2008*) between climate model grid cell centroids. Climate simulations for each month were then created by applying interpolated delta values to each observed grid cell value.

To maintain a range of variability in climate projections while reducing computation time, we selected 7 of the 70 available climate simulations as drivers (Table S1), using an objective approach that employs a k-means clustering approach to obtain a good coverage of overall future uncertainty (*Casajus et al.*, *2016*). We considered all selected scenarios as equiprobable in this analysis.

Note that emission scenarios are now represented by four Representative Concentration Pathways (RCPs), which became available with the IPCC fifth assessment report and after this study was initiated. The RCPs span a larger range of stabilization, mitigation and non-mitigation pathways than the range covered by the SRES we used (Table S2). As a result, the RCPs now estimate a larger range of temperature increase than the SRES (*Rogelj, Meinshausen & Knutti*, *2012*). Moreover, climate models have also been developed since phase 3 of the Coupled Model Intercomparison Project—CMIP3, particularly by including the representation of biogeochemcal cycles (*Flato et al.*, *2013*). Some models do perform better than others for certain climate variables, but no individual model is clearly the best overall. The comparison of median model capability in reproducing historical climate shows relatively modest improvement between CMIP3 and the current CMIP5 generation (*Flato et al.*, *2013*). Each generation exhibits a range in performance, with CMIP5 showing fewer 'bad models' than CMIP3, but the species distribution modelling uses a consensus approach similar to looking at an ensemble mean of General Circulation Models. As well, for climate projections, we used the delta method (see above) where only relative changes from the GCMs are calculated and then applied to observed data. In this case, a similar range in changes of temperature and precipitation between CMIP5 and CMIP3 is what matters most in terms of species distribution model results. A detailed comparison of CMIP3 and CMIP5 is beyond the scope of this paper, but interested readers can consult *Flato et al.* (*2013*), particularly their Figs. 9.44 and Fig. 1 of FAQ 9.1.

### Topographic and soil data

Elevation data were provided by the Climate Change Tree Atlas database for the eastern United States portion of the modelling area (*Landscape Change Research Group*, *2014*), whereas for Quebec it was obtained from the Canadian Surface Model Mosaic (http://geogratis.gc.ca/api/en/nrcan-rncan/ess-sst/3A537B2D-7058-FCED-8D0B-76452EC9D01F.html) at a resolution of ca. 20 m and averaged to match our grid. For the eastern United States part of the modelling area, we obtained soil characteristics data (surface deposit and drainage class; Table 1) from the NRCS Soil Survey data (version 2.1, scale 1:24,000; http://websoilsurvey.nrcs.usda.gov/). For the Quebec part, we used soil data from the 3rd decennial permanent and temporary surveys (1:20 000 scale) of the ministère des Forêts, de la Faune et des Parcs, Quebec (available on request at http://www.mffp.gouv.qc.ca/forets/inventaire/donnees-inventaire.jsp). For each grid cell, we computed the percentage of the 20-km × 20-km cell occupied by each level of each edaphic variable.

## Modelling current and future habitat suitability
### Species distribution modelling

We computed the geographical distribution of suitable climatic and edaphic conditions—or habitat, as defined by these particular dimensions of the niche—for each of the target tree species, following an ensemble procedure (*Araújo & New*, *2007*) with the BIOMOD 1.1 modelling package (*Thuiller et al.*, *2009*) implemented in R (*R Development Core Team*, *2010*). We considered both a baseline period (1961–1990) and a future period (2071–2100, hereafter referred to as 2080).

We used species occurrence data and environmental predictors to build species distribution models using eight modelling techniques: three regression methods (generalized additive models, GAM; generalized linear models, GLM; multivariate adaptive regression splines, MARS), two classification methods (mixture discriminant analysis, MDA; classification tree analysis, CTA) and three machine learning methods (artificial neural networks, ANN; generalized boosted models, GBM; random forest, RF). All models were produced using default BIOMOD parameters where appropriate (*Thuiller et al.*, *2009*). Further parameters were as follows: GLMs were generated using quadratic terms and a stepwise procedure with the AIC criteria; GAMs were generated with a spline function with three degrees of smoothing; GBMs were built with a maximum of 2,000 trees; ANNs were produced with five cross-validations (see *Marmion et al.*, *2009a* for further details on these modelling techniques). For each species, we built the eight species distribution models using a random subset of data containing 70% of the 20 × 20-km cells (i.e., 4,493 cells). We used the remaining 30% (i.e., 1,925 cells) to evaluate the predictive performance of the models. We repeated this split-sample procedure ten times, thus calibrating 80 different statistical models for each species. We simulated suitability under climate change (future suitability) by projecting each of the 80 projections under each of the seven climate simulations for 2080. This generated a total of 560 probabilities (ten repetitions × eight modelling techniques × seven climate simulations) of habitat suitability for each species for the 2080 period. We combined the different probabilities of habitat suitability (P) based on the area

under the receiver-operating characteristic (ROC) curve (AUC) values; we assigned the AUC values from each modelling technique as the weights of the weighted average in order to enhance the contributions of models with higher performance values:

$$\text{WAP}_{i_{\text{baseline}}} = \frac{\sum_{j=1}^{8} \sum_{k=1}^{10} \left( \text{AUC}_{jk} \times P_{ijk} \right)}{\sum_{j=1}^{8} \sum_{k=1}^{10} \left( \text{AUC}_{jk} \right)} \tag{1}$$

$$\text{WAP}_{i_{2080}} = \frac{\sum_{j=1}^{8} \sum_{k=1}^{10} \sum_{l=1}^{7} \left( \text{AUC}_{jk} \times P_{ijkl} \right)}{7 \times \sum_{j=1}^{8} \sum_{k=1}^{10} \left( \text{AUC}_{jk} \right)} \tag{2}$$

where WAP is the weighted average probability of habitat suitability, $i$ is the index of the grid cell (1, …, 6418), $j$ is the modelling technique (GAM, GLM, MARS, CTA, MDA, ANN, GBM, RF), $k$ is the repetition (1, …, 10) and $l$ is the climate simulation (1, …, 7). Averaged projections resulted in a single projection at each grid cell for each species (hereafter referred as the "average model") for the baseline period (WAP$_{i_{\text{baseline}}}$; Eq. 1) and the 2080 period (WAP$_{i_{2080}}$; Eq. 2). This method is considered to be more robust than other model fusion methods or single model projections (*Marmion et al., 2009b*).

### Transforming probabilities of suitability to binary values

To transform continuous probabilities of suitability into binary (0/1) values, we calculated a common threshold (cut-off) value for both the baseline period and the 2080 period using a binary vector of observed occurrence and a vector of probability of occurrence from the average model (WAP$_{i_{2080}}$). We searched for the threshold which jointly maximized sensitivity and specificity (*Liu et al., 2005*). This approach is considered among the most reliable for choosing a threshold (*Freeman & Moisen, 2008*).

### Model evaluation

The predictive model performance was evaluated using area under the receiver operating characteristic curve (AUC; *Fielding & Bell, 1997*) as an accuracy measure. The area under the ROC function (AUC) is usually taken to be an important index because it provides a single measure of overall accuracy that is not dependent upon a particular threshold. Suggested AUC values for classifying the accuracy of models using AUC are: 0.90–1.00 = excellent; 0.80–0.9 = good; 0.70–0.80 = fair; 0.60–0.70 = poor; 0.50–0.60 = fail (e.g., *Virkkala et al., 2010* adapted from *Swets, 1988*). Sensitivity (true positive fraction) and specificity (false positive fraction) values were also reported for each species (*Lobo, Jiménez-Valverde & Real, 2008*).

### Agreement between the average future projection in each cell and the single projections

To measure the level of confidence in our average future projection for a given cell, we also calculated the percentage of the 560 single projections for that cell that agreed with the average projection (hereafter referred as "agreement value").

### Identifying vulnerable habitats under future climates

We focused on Quebec's productive forest territory to evaluate whether predicted future conditions remained suitable for a species within its baseline range. For this
purpose, the baseline range of a species was defined as the set of grid cells within Quebec productive forests where the baseline average model predicted a suitable habitat ($WAP_{i_{baseline}} \geq$ threshold value), as defined by climatic, edaphic and topographic variables. Note that a 'suitable habitat' does not necessarily mean an 'optimal habitat', since a species can be found on sites with suboptimal conditions. Cells modelled as suitable habitat under baseline climatic conditions, but which became unsuitable under future climate conditions, were classified as unsuitable habitat (UH). Cells modelled as 'suitable' under both baseline and future climate further subdivided as:

Less Suitable Habitat (LSH):

$$\left[ WAP_{i_{2080}} - WAP_{i_{baseline}} < 0 \&, \left| WAP_{i_{2080}} - WAP_{i_{baseline}} \right| \geq 0.15 \right] \tag{3}$$

Persistent Habitats (PH):

$$\left[ WAP_{i_{2080}} - WAP_{i_{baseline}} < 0 \& \left| WAP_{i_{2080}} - WAP_{i_{baseline}} \right| < 0.15 \right] \tag{4}$$

OR

$$\left[ WAP_{i_{2080}} - WAP_{i_{baseline}} \geq 0 \right]. \tag{5}$$

LSH reflects predicted probabilities of habitat suitability that decrease over time, but not to the point of unsuitability like UH. We used the threshold of a 15% change of probabilities of habitat suitability ($WAP_{i_{2080}} - WAP_{i_{baseline}}$) to select the proper subcategory for each cell. This threshold was chosen after examining spatial predictions for 2050 and comparing them with predictions for 2080. The majority of cells classified as less suitable at the 15% threshold in 2050 became unsuitable in 2080. A sensitivity analysis, where the threshold value varied from 5% to 25%, showed how forecasts are affected when this value changes (Fig. S7).

For each species, we reported trends in relation to the entire productive forest territory, the baseline range of the species in Quebec, and each of five vegetation domains.

## RESULTS

### Model evaluation

Overall, all the models performed well and showed good capacity on species prediction as accuracies showed high values (Table S3). The AUC values of the average models ranged from 0.916 (sugar maple) to 0.984 (for balsam fir), for a mean value of 0.958 ± 0.029. We also determined that the largest part of variability in future projections was explained by the SDMs (Fig. S8).

### Assessing risk under future climate

Species are presented in order of decreasing importance in the study area (as measured by size of their baseline range in Quebec's productive forest).

#### Black spruce (Table 2; Fig. 2A)

The baseline range for black spruce in the study area essentially covers all five bioclimatic domains. Overall, 78% of the baseline range of black spruce in Quebec's productive forest

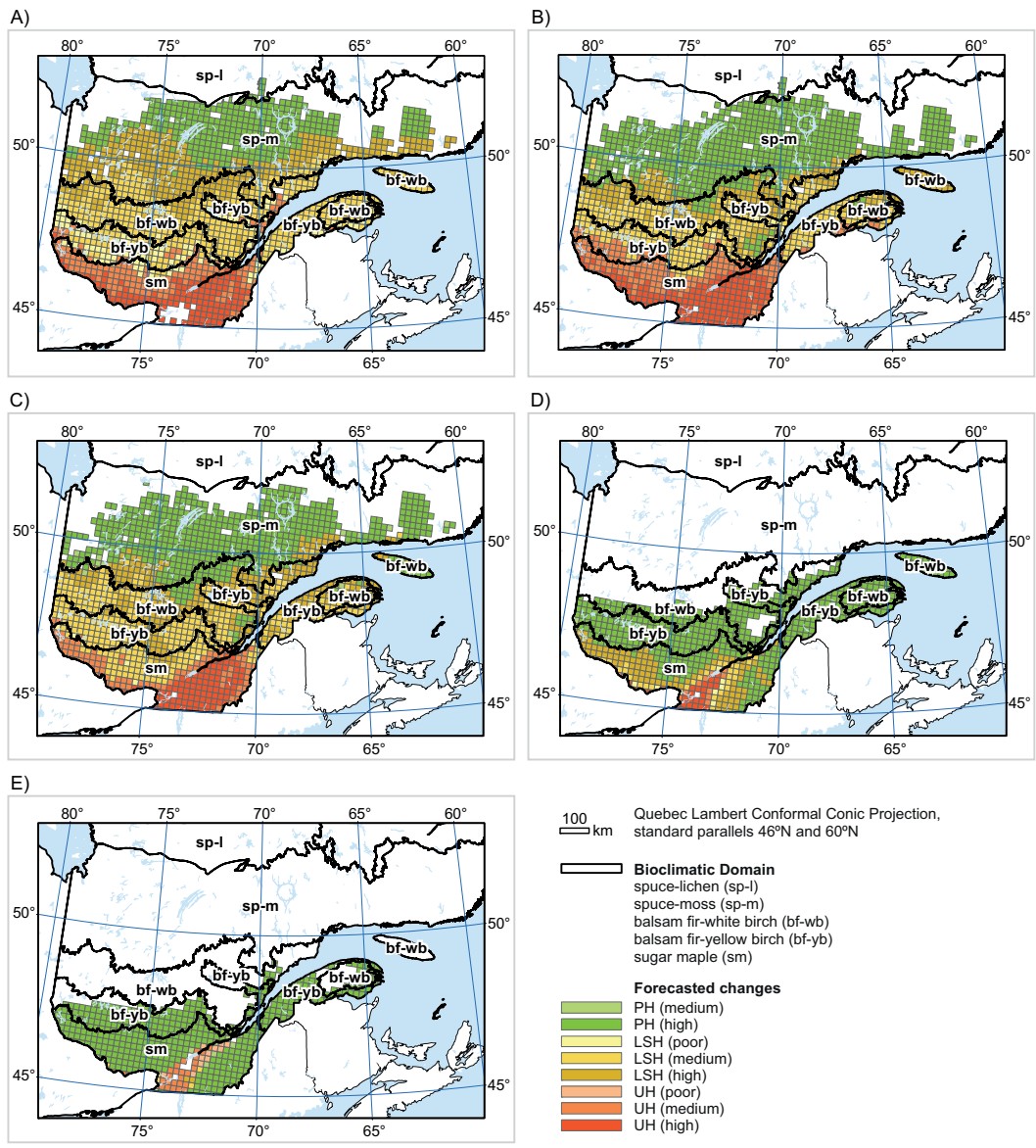

**Figure 2** **Forecasted changes (2080) in (A) black spruce habitat, (B) balsam fir habitat, (C) white birch habitat, (D) yellow birch habitat and (E) sugar maple habitat.** UH: unsuitable habitat; LSH: persistent but less suitable habitat; PH: persistent habitat. Confidence values were calculated as the percentage of the 560 single predictions for a given cell that agreed with the average prediction for that cell. Values ≤ 50%: poor; 50% < values ≤ 75%: medium; values > 75%: high.

is projected to shift towards unsuitable (18%) or less suitable (60%) conditions compared to baseline conditions (agreement value = 68%). Shifts in suitability are projected largely within the sugar maple domain (89% of baseline spruce habitat in that domain shifting to unsuitable), the balsam fir–yellow birch domain (13% shifting to unsuitable), and the balsam fir–white birch domain (2% shifting to unsuitable). Moreover, all the remaining baseline habitats in these domains are projected to become less suitable for black spruce

compared to baseline climatic conditions. In the spruce–moss domain, 52% of suitable habitats are projected to become less suitable for the species.

### Balsam fir (*Table 2*; *Fig. 2B*)

The baseline range for balsam fir covers more than 97% of Quebec's productive forests. Overall, 59% of the baseline range of balsam fir is projected to shift towards unsuitable (21%) or less suitable (38%) climatic conditions (agreement value = 69%) with climate change. Essentially, all baseline sites over the entire sugar maple domains and the balsam fir–yellow birch domain are projected to become unsuitable or less suitable. Further north, in the balsam fir–white birch domain, shifts towards unsuitability are projected on 1% of the range, while less suitable conditions are projected on another 87%.

### White birch (*Table 2*; *Fig. 2C*)

White birch is widely distributed in the study area with a baseline range covering 94% of Quebec's productive forests. Overall, 62% of the baseline range of balsam fir is projected to shift towards unsuitable (14%) or less suitable (48%) climatic conditions (agreement value = 71%) with climate change. In the sugar maple domains, unsuitability is projected on 63% of the baseline range, with the remainder projected as less suitable compared to baseline conditions. Only 2% of habitats shifts towards unsuitability in the balsam fir–yellow birch domain, but less suitable habitats are projected in 67% of the balsam fir–yellow birch domain, 79% of the balsam fir–white birch domain, and 15% of the spruce–moss domain.

### Yellow birch (*Table 2*; *Fig. 2D*)

The baseline range for yellow birch covers 44% of Quebec's productive forests. Shifts towards unsuitability (5%) or less suitability (19%) are projected over 24% of the baseline range (agreement value = 78%). All unsuitable areas are in the sugar maple domains (13%), as are most habitats projected as less suitable (48%).

### Sugar maple (*Table 2*; *Fig. 2E*)

The baseline range of sugar maple covers 31% of Quebec's productive forests, essentially in the south. Shifts towards unsuitability (8%) or less suitability (1.3%) are projected for 9.3% of the sugar maple baseline range (agreement value = 60%). All sites shifting to unsuitable conditions are in the sugar maple domains. The more northern domains are predicted to maintain their current habitats for sugar maple.

## DISCUSSION

Most studies linking climate change with species distribution models emphasize the potential for major shifts in species ranges and a massive reorganisation of biodiversity. Our study is no exception but here we focus on areas where species are projected to become at risk of climate change-related stress to help define adaptation strategies. We define 'risk' as a function of the probability of an event (climate becoming unsuitable or less suitable for a species as projected) and the severity of its consequences (*FAO*, *2007*; *Leung et al.*, *2012*). Whereas one can rightly argue that there is still much uncertainty in assessing probability of

**Table 2  Impact of climate change on tree habitat suitability in 2080.** Forecasted changes in species habitat are illustrated both as absolute areas (km$^2$) and proportion of the baseline range for the region (% of baseline). The baseline (1961–1990) range of a species is the total area (km$^2$) of all cells where the baseline average model predicted a suitable habitat for that species, within each bioclimatic domain or for all of the Quebec productive forest. The average agreement (% ag.) was calculated as the mean percentage, within a given region, of single predictions for a given cell that agreed with the average prediction for that cell.

| Species/Region | Baseline range (km$^2$) | Forecasted changes in species habitat | | | | | | | | |
| --- | --- | --- | --- | --- | --- | --- | --- | --- | --- | --- |
| | | Unsuitable habitat | | | Less suitable habitat | | | Persistent habitat | | |
| | | km$^2$ | % of baseline | % ag. | km$^2$ | % of baseline | % ag. | km$^2$ | % of baseline | % ag. |
| **Black Spruce** | | | | | | | | | | |
| Sugar maple domain | 103,570 | 92,348 | 89 | 77 | 11,222 | 11 | 48 | | | |
| Balsam fir–yellow birch domain | 97,152 | 12,347 | 13 | 66 | 84,804 | 87 | 55 | | | |
| Balsam fir–white birch domain | 136,977 | 2,778 | 2 | 65 | 134,199 | 98 | 65 | | | |
| Spruce–moss domain | 268,668 | 209 | <0.1 | 66 | 139,505 | 52 | 76 | 128,953 | 48 | 94 |
| Spruce–lichen domain | 2,660 | | | | | | | 2,660 | 100 | 99 |
| Total (Quebec productive forest) | **609,027** | **107,682** | **18** | **74** | **369,730** | **61** | **65** | **131,614** | **21** | **94** |
| **Balsam fir** | | | | | | | | | | |
| Sugar maple domain | 109,063 | 103,583 | 95 | 76 | 5,481 | 5 | 57 | | | |
| Balsam fir–yellow birch domain | 97,152 | 21,249 | 22 | 59 | 75,897 | 78 | 63 | 6 | <0.01 | 84 |
| Balsam fir–white birch domain | 136,977 | 1,262 | 1 | 58 | 118,967 | 87 | 70 | 16,748 | 12 | 89 |
| Spruce–moss domain | 253,288 | | | | 29,929 | 12 | 77 | 223,359 | 88 | 90 |
| Spruce–lichen domain | 2,563 | | | | | | | 2,563 | 100 | 99 |
| Total (Quebec productive forest) | **599,042** | **126,093** | **21** | **71** | **230,273** | **38** | **68** | **242,675** | **41** | **90** |
| **White Birch** | | | | | | | | | | |
| Sugar maple domain | 109,077 | 69,167 | 63 | 75 | 39,910 | 37 | 63 | | | |
| Balsam fir–yellow birch domain | 97,152 | 1,846 | 2 | 61 | 94,926 | 98 | 69 | 379 | | 86 |
| Balsam fir–white birch domain | 136,977 | 5 | <0.01 | 59 | 108,101 | 79 | 72 | 28,871 | | 86 |
| Spruce–moss domain | 235,395 | | | | 34,631 | 15 | 79 | 200,764 | 85 | 85 |
| Spruce–lichen domain | | | | | | | | | | |
| Total (Quebec productive forest) | **578,600** | **71,019** | **12** | **74** | **277,568** | **48** | **70** | **230,014** | **40** | **85** |
| **Yellow Birch** | | | | | | | | | | |
| Sugar maple domain | 109,077 | 13,915 | 13 | 79 | 52,434 | 48 | 77 | 42,728 | 39 | 95 |
| Balsam fir–yellow birch domain | 95,316 | | | | 372 | < | 87 | 94,944 | 99 < | 93 |
| Balsam fir–white birch domain | 66,705 | | | | | | | 66,705 | 100 | 94 |
| Spruce–moss domain | 1,469 | | | | | | | 1,469 | 100 | 100 |
| Spruce–lichen domain | | | | | | | | | | |
| Total (Quebec productive forest) | **272,567** | **13,915** | **5** | **79** | **52,806** | **19** | **78** | **205,847** | **76** | **94** |

**Table 2** (*continued*)

| Species/Region | Baseline range (km²) | Forecasted changes in species habitat | | | | | | | | |
|---|---|---|---|---|---|---|---|---|---|---|
| | | Unsuitable habitat | | | Less suitable habitat | | | Persistent habitat | | |
| | | km² | % of baseline | % ag. | km² | % of baseline | % ag. | km² | % of baseline | % ag. |
| **Sugar Maple** | | | | | | | | | | |
| Sugar maple domain | 106,902 | 14,375 | 13 | 57 | 2,536 | 2 | 71 | 89,990 | 84 | 89 |
| Balsam fir–yellow birch domain | 69,917 | | | | | | | 69,917 | 100 | 94 |
| Balsam fir–white birch domain | 11,683 | | | | | | | 11,683 | 100 | 95 |
| Spruce–moss domain | 209 | | | | | | | 209 | 100 | 99 |
| Spruce–lichen domain | | | | | | | | | | |
| **Total (Quebec productive forest)** | **188,712** | **14,375** | | 57 | **2536** | | 71 | **171,800** | | 92 |

species occurrence in a changing climate, there is no doubt that the consequences of habitat decline at a particular location can be highly significant for ecosystems and economies that depend on, or are defined by these species. Risk assessment through climate/species models, therefore, has at least two immediate benefits. Just as for climate projections, it can help draw attention of policy makers, forest management agencies, and the public in general on the sheer magnitude of projected climate change effects on biodiversity. Secondly, because models are spatially-explicit and species-specific, they can help target monitoring efforts, especially when resources are scarce, and potentially inform adaptation strategies.

The consequences of an unsuitable climate on species can be associated with a range of processes directly or indirectly related to climate change, including increased physiological stress induced by heat or drought (*Anderegg et al.*, *2015*; *Park Williams et al.*, *2013*; *Sun et al.*, *2015*; *Wu et al.*, *2012*), increased vulnerability to pest and disease outbreaks (*Creeden, Hicke & Buotte*, *2014*; *DeRose et al.*, *2013*; *Fierravanti et al.*, *2015*), competition from other species (*Blois et al.*, *2013*; *Brooker*, *2006*; *Carón et al.*, *2015*; *Dukes et al.*, *2009*; *Meier et al.*, *2012*) or herbivory (*Svenning & Sandel*, *2013*), and increased climate-mediated frequency of fires or destructive weather events (*Bergeron et al.*, *2010*; *Terrier et al.*, *2013*). However, the precise pathways through which climate change will affect a particular forest remains difficult to predict, as is the attribution of any particular event to climate change. Based on the proportion of their baseline range that is projected to become unsuitable, our target species rank as follows, in decreasing order of vulnerability: balsam fir (21%), black spruce (18%), white birch (14%), sugar maple (8%), and yellow birch (5%). In the 185,000-km$^2$ area where the baseline ranges of all five species intersect, at least three species—and, in the southernmost part of the study area, all five of them—are projected to be at some risk of climate-related stress (Fig. 3). This represents a significant proportion of Quebec forests and suggests that species that have traditionally defined whole regional vegetation assemblages could become less characteristic of these regions. Forest decline would have, as well, consequences on the value of forest land (*Hanewinkel et al.*, *2012*).

Because of the strong north–south climatic gradient in Quebec, species are projected to retract from their southern margins in the study area with warming. Biotic interactions are often emphasized over climate in determining southern range edges (*Normand et al.*, *2009*; *Sunday, Bates & Dulvy*, *2012*), and so this raises the question of whether competitive processes mediated by species traits over novel climatic conditions will help shift dominance of species locally. For instance, balsam fir is more fire-sensitive than black spruce and shifts in fire regime in the northern boreal forests over millennia have shifted dominance towards one species or the other, with warm and wet conditions favouring balsam fir over black spruce (*Ali et al.*, *2008*; *Couillard, Payette & Grondin*, *2013*). Moreover, the observed northward migration of pests, such as spruce budworms, facilitated by climate change is also contributing to increase the intensity and frequency of outbreaks in some areas. Whereas balsam fir is currently considered a more suitable host than black spruce, this may change when the budworm hits spruce-dominated forests (*Pureswaran et al.*, *2015*).

Warming experiments can show direct physiological effect on individual trees, but it is not always clear how warming can influence whole species assemblages over a range of soil conditions. Increased frequency and intensity of droughts, for instance, have led to negative

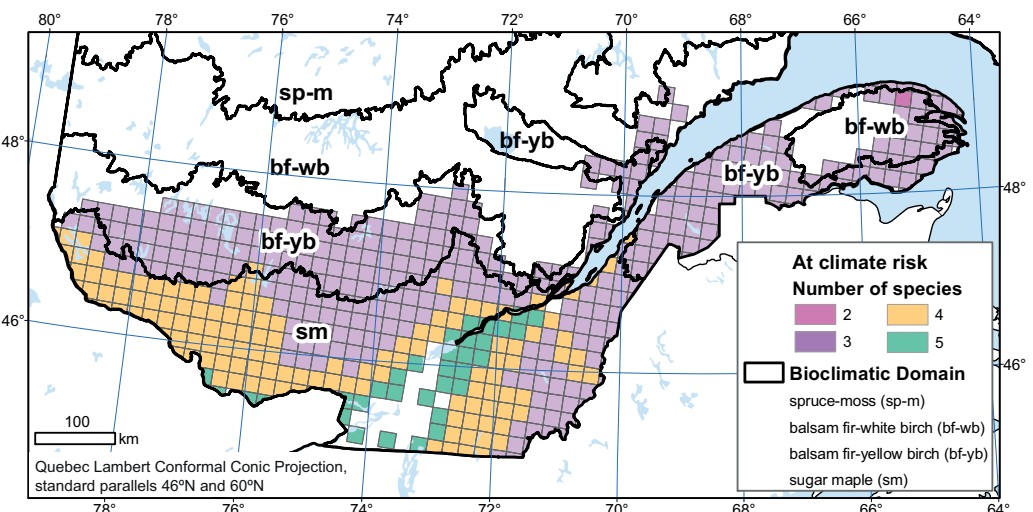

**Figure 3 Number of tree species, among the 5 studied species, at risk of some climate-related stress in 2080.** We considered only cells in the study area where the baseline habitat was suitable for all 5 species.

effects on the duration of xylogenesis and the production of xylem cells in balsam fir in warming experiments (*D'Orangeville et al.*, *2013*). For sugar maple, the observation that adverse winter and spring conditions in southern sites negatively impact maple syrup production may provide early indication for warming effect (*Duchesne & Houle*, *2014*). As decreasing growth rates can precede mortality, an even stronger signal comes from the observation of widespread decreasing growth rate for sugar maple documented in the Adirondacks (*Bishop et al.*, *2015*). While underlying mechanisms have to be clarified, these observations are in agreement with niche model projections in the eastern US (*Iverson et al.*, *2008*).

There is uncertainty in model projections because of uncertainty in climate simulations, statistical models, and the non-linear responses of ecosystems and species. Climate simulations are improving (*Flato et al.*, *2013*) and the limitations of different statistical models are well recognised (*Marmion et al.*, *2009a*). These limitations are often taken into account, for instance by using consensus approaches across several statistical and climate models (*Guo et al.*, *2015*; *Wang et al.*, *2016*). The level of agreement among our projections was generally high (averaging 75%), raising confidence in our results given the data available. Nevertheless, the main source of uncertainty may rest not so much in the methodology used than in the model assumptions. There is no doubt that climate is a strong predictor of site occupancy patterns for species, particularly at broad spatial scale (*Pearson & Dawson*, *2003*). What remains unclear, however, is the extent to which climate mainly determines species range boundaries and whether current distribution patterns really capture the physiological limits of species (*Brown & Vellend*, *2014*; *García-Valdés et al.*, *2015*; *Nowacki & Abrams*, *2015*; *Paul, Bergeron & Tremblay*, *2014*). The availability of suitable conditions other than climate (*Beauregard & De Blois*, *in press*), postglacial dispersal limitations, or competition can all contribute to species not filling their available climatic niche (*Sinclair, White & Newell*, *2010*). Coupling physiological models or trait information with correlative range models can help refine projections (*Iverson et al.*, *2011*;

*Talluto et al.*, *2016*), providing that physiological models capture species responses outside the range of conditions represented by species presence-absence data. If there is, for instance, evidence for climatically suitable sites colder than those currently captured by the observed species' range, the consequences may be minimal on risk assessment related to warming. If, on the other hand, there is evidence for climatically suitable sites warmer than those currently defined by a species' range—or greater tolerance to warming than previously thought , future projections are likely to overestimate the risk of climate change on species distribution. Since species interactions also influence species distribution—but are somewhat integrated in models based on a species' realised niche, another unresolved issue is how communities will reassemble. Disagreements as to the geographical extent of climate vulnerability are likely to persist until monitoring and field evidence clearly show trends in support of (or in disagreement with) projections in a given region. Models can only point towards species or areas at risk for greater scrutiny and, most of all, provide incentive for developing and testing adaptation strategies.

If projections in this study question the future relevance of the current ecological classification of the forest landscape, they also raise important issues regarding the forest management regime, especially under the assumption that an ecosystem is defined by a relatively stable climate and substrate. The ecological principles that underlie current ecosystem-based management emphasize the need to reduce the differences between natural and managed landscapes (*Gauthier et al.*, *2009*). They imply that sustainable forest management practices should aim for a desired composition and age structure. This becomes quite a challenge if the target composition is moving fast under a new climate regime (*Dhital et al.*, *2015*; *Mori et al.*, *2013*). Therefore, the greatest challenges in coming years will be to manage rapid transitions of forests towards other, largely unknown, steady-states. As a result, the adaptation literature has repeatedly highlighted the need to move from a paradigm of preserving current conditions or restoring 'historical fidelity' to one of managing for novel ecosystems that may differ in composition, structure, and/or function (*Hobbs, Higgs & Harris*, *2009*). Models provide some indications of where the challenges could be the greatest, and whether or not species at risk are worth maintaining at specific locations under a shifting climate. Publicly managed forests in the study area, for instance, are restored to production largely by prioritizing practices that protect the established regeneration. Where regeneration is insufficient, as may increasingly be the case on sites that we identified as 'at risk', reforestation may be carried out. However, the choice of species is for the most part still made under the assumption that suitable conditions in this century will be similar to the ones in recent history. New practices are being tested to maximize forest resiliency while taking into account transition states, for instance by helping shift composition (including genetic variability) towards species or individuals adapted to the new climate regime (*Breed et al.*, *2012*; *Koralewski et al.*, *2015*; *Park Williams et al.*, *2014*). As well, maintaining biodiverse (both in terms of composition and age structure) forests and landscapes could provide some insurance against instability (*Churchill et al.*, *2013*; *Thompson et al.*, *2009*).

Our study area covers large regions where forest logging, especially of softwood stands, contributes significantly to the economy. Forests provide habitats and contribute to

global carbon storage. Be it with species distribution models (*Hufnagel & Garamvolgyi, 2014*), more detailed process-based models (*Zolkos et al., 2014*), warming experiments (*Dulamsuren et al., 2013*) or field evidence (*Dudley, Burns & Jacobi, 2015; Girardin et al., 2014; Worrall et al., 2013*), all attempts to translate climate simulations into forest patterns converge towards the same message: trees could be at risk of maladaptation over a remarkably large proportion of their baseline range in this century. Sustaining yield could become increasingly difficult in these conditions. Reforestation planning will have to take into account climate trajectory and maps indicating areas at risk. Although it will be tempting to log declining forests, it will be as important to preserve reference areas under natural disturbances in order to understand 'natural' dynamics and adapt management options accordingly. New engagement rules with the forest industry, which may see areas at risk as opportunities for salvage logging, will be needed.

To respond to the climate change challenge for forests, efforts are focusing on three fronts: (1) Risk assessment, including the targeted monitoring of areas at risk, in order to understand forest dynamics under changing conditions. Quebec has the advantage of having established a large network of forest sites under observation since the 1970s (*Ministère des Forêts de la Faune et des Parcs, 2014*). Assessment of climate change-related risk is probably where most research efforts have focused so far, but there is still a need to better identify and target areas and species at risk. (2) Risk communication with stakeholders, decision makers, and the public in general. During the last decade, a great deal of work has been done to provide conceptual frameworks and provide new approaches and tools for decision making under uncertainty (*Janowiak et al., 2014*). In Quebec, the recent publication of the results of a large study involving scientists and stakeholders on the impacts of climate change on Quebec biodiversity is a positive step in that direction (*Berteaux, Casajus & De Blois, 2014*). When communicating risk, it is indeed important to indicate the uncertainty inherent in all projections. However, the scientific emphasis on uncertainty has also been seen as possibly deterring from early action regarding climate change policies (*Morton et al., 2011*). (3) Risk management, which involves basing decisions on the best information available (*Yousefpour et al., 2014*). This may be the most challenging aspect. Comparing the outcomes of alternative management scenarios in relation to predicted responses of forest to climate change could inform management decisions (*Messier et al., 2016*). If they are not already in place, adaptation strategies are needed, if only to allow sufficient time for forest ecosystems and regional forest economies to adapt. In spite of their well-recognised limitations, regionally-explicit risk assessment approaches, such as the one used here, currently remain one of the best options to convey the need for climate policies and forest management adaptation strategies.

## ACKNOWLEDGEMENTS

The authors thank Travis Logan and Ouranos for providing the climate-scenario data, Marie-Claude Lambert and Nicolas Casajus for assisting with statistical analyses, Denise Tousignant for scientific edition, and reviewers for their helpful suggestions.

### Funding

Funding for this research was provided by the ministère des Forêts, de la Faune et des Parcs du Québec and the Government of Quebec's Green Fund within the framework of measure 24 of the 2006–2012 Climate Change Action Plan and research was conducted as part of the DRF's project 142332119. The funders had no role in study design, data analysis, decision to publish, or preparation of the manuscript. Data of Quebec were provided by the ministère des Forêts, de la Faune et des Parcs, through forest inventories.

### Grant Disclosures

The following grant information was disclosed by the authors:
ministère des Forêts, de la Faune et des Parcs du Québec.
Government of Quebec's Green Fund.
DRF's project: 142332119.

### Competing Interests

The authors declare there are no competing interests.

### Author Contributions

- Catherine Périé conceived and designed the experiments, performed the experiments, analyzed the data, contributed reagents/materials/analysis tools, wrote the paper, prepared figures and/or tables, reviewed drafts of the paper.
- Sylvie de Blois conceived and designed the experiments, performed the experiments, contributed reagents/materials/analysis tools, wrote the paper, reviewed drafts of the paper.

### Data Availability

Raw data and R code are provided as Supplemental Information 1 and 2.

### Supplemental Information

Supplemental information for this article can be found online at http://dx.doi.org/10.7717/peerj.2218#supplemental-information.

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
