# Peer review of "Dominant forest tree species are potentially vulnerable to climate change over large portions of their range even at high latitudes"

_PeerJ, doi:10.7717/peerj.2218_

## Round 0.1 · original submission · Major Revisions

I enjoyed reading the paper and thought it made a useful contribution to the literature examining forest vulnerability to climate change. In general I found it to be well-written and carefully thought through. I appreciated the care with which the authors utilised and interpreted the bioclimatic niche modeling approach given existing concerns about its limitations. I think all the reviewers were in agreement that the paper is suitablie for publication. There are, however, five key areas that need your attention and which may warrant a bit more work:

1) Reviewer 2 expressed concern that the 15% threshold level you utilize needs some attention. This is something I noted in my own review. Reviewer 2 suggests you present a sensitivity analysis to examine how your forecasts change with alteration to this value - I think that would be a useful addition

2) Reviewer 3 expresses some concerns about the spatial scale of your modeling work. It would be good to see you justify this in your paper

3) Reviewers 1 and 3 draws attention to the fact that more up-to-date GCMs and climate scenarios are available and question why you didn't choose to use them

4) Reviewer 3 and I were concerned about the potential for colinearity between your predictor variables - I realize that you addressed this for the climate variables but it would be good to see you address the potential for colinearity between climate and other environmental variables (e.g. elevation, soil type).

5) I thought it was interesting that you discussed model uncertainty a fair amount but did not quantify what the variability between your model climate scenarios and modeling methods looked like (though I recognize you did include the agreement analysis). Might that make an interesting addition?

I look forward to seeing a revised version of this manuscript. I've attached a pdf copy in which I've made a number of comments and minor corrections.

·

Basic reporting

No comments

Experimental design

My only major concern is the choice of climate models: all those used seem to be more than 10 years old. It is not just the emissions scenarios that have changed since then. There have also been major changes in the models themselves. The consequences for climate projections vary from place to place, but are sometimes large. The authors need to explain and justify these choices and at least briefly compare the climate projections they use with those produced by the CMIP5 ensemble.

Validity of the findings

The findings appear to be valid for the models used.

Reviewer 2 ·

Basic reporting

In terms of basic reporting, I found the manuscript to be clear and concise, with sufficient information provided in the supplemental materials. The purpose and rationale for the study is clear from the Introduction, and the methods are organized and presented in a clear and concise fashion. I found the figures and tables to be appropriate and necessary.

Experimental design

Not being an ecological modeler, I cannot comfortably claim that I am an expert in the specifics of the models and design. I was able to follow the methods and understand the approach taken at a broad scale, but I cannot comment on the appropriateness of the modeling approach and linkage between the species distribution and climate models. The literature cited appears to include the relevant material I am familiar with, including the work by Louis Iverson and the USFS Northern Research Station. The statistical approach to examining the scenarios seems appropriate (including the bootstrap method), however, more information on the specific methods could be incorporated (see cluster analysis). Finally, I think that a sensitivity analysis of the 15% threshold value for determining a shift in habitat should be conducted to help interpret the long-term changes in forest species composition.

Validity of the findings

As for the validity of the findings, I believe they add to our understanding of potential climate change impacts of forest distributions. The authors correctly explore the implications of the findings in the Discussion, however, I found the statements in the final paragraph to be excessive arm-waving. I would suggest this section be revised and the tone of the recommendations be more objective. Furthermore, I expected to see more about how other factors driven by climate such as wildland fire or insect attacks would potential impact species distributions, especially in the light of the authors speaking to “novel” ecosystems (a consequence I happen to agree with the authors about). This aspect should be incorporated in the manuscript.

Additional comments

This is a generally well-written manuscript that explores the potential distribution of major forest species in Quebec under a changing climate. The authors link an extensive dataset of tree distributions from the U.S. and eastern Canada with climate change models to predict the relative change in 20 km by 20 km cells among current forest types over the next 100 years. After simulating how these species distributions will change over time and comparing different scenarios, the authors show that all five major species that are important to forest industry sector of Quebec will experience shifts in their distribution, primarily declines in suitable habitat. The authors conclude by providing some implications of their findings to the broader forestry sector of Quebec, and suggest a call-for-action to help mitigate these impacts.

In terms of basic reporting, I found the manuscript to be clear and concise, with sufficient information provided in the supplemental materials. The purpose and rationale for the study is clear from the Introduction, and the methods are organized and presented in a clear and concise fashion. I found the figures and tables to be appropriate and necessary.

Not being an ecological modeler, I cannot comfortably claim that I am an expert in the specifics of the models and design. I was able to follow the methods and understand the approach taken at a broad scale, but I cannot comment on the appropriateness of the modeling approach and linkage between the species distribution and climate models. The literature cited appears to include the relevant material I am familiar with, including the work by Louis Iverson and the USFS Northern Research Station. The statistical approach to examining the scenarios seems appropriate (including the bootstrap method), however, more information on the specific methods could be incorporated (see cluster analysis). Finally, I think that a sensitivity analysis of the 15% threshold value for determining a shift in habitat should be conducted to help interpret the long-term changes in forest species composition.

As for the validity of the findings, I believe they add to our understanding of potential climate change impacts of forest distributions. The authors correctly explore the implications of the findings in the Discussion, however, I found the statements in the final paragraph to be excessive arm-waving. I would suggest this section be revised and the tone of the recommendations be more objective. Furthermore, I expected to see more about how other factors driven by climate such as wildland fire or insect attacks would potential impact species distributions, especially in the light of the authors speaking to “novel” ecosystems (a consequence I happen to agree with the authors about). This aspect should be incorporated in the manuscript.

Additional comments:

L44-45: Citation for this idea that climate is considered “stable”? I agree and think the authors could look to the ecological classification work in the US or habitat typing to provide support for this concept.

L51: Would suggest replacing “will” with “may.”

L85: insert “the” before “eastern United States…” and no hyphen needed.

L107: (and elsewhere) should “minstere” be capitalized?

L162: Capitalize “department of forests, wildlife, and parks”?

L173-174: spell out numbers (e.g., three, one, seven, and three)

L181: I don’t understand why only three variables were used out of the 35; more detail on why is needed beyond just multicollinearity. How was multicollinearity determined? Could a PCA be used to summarize the climate variables to extract the dominant variation in climate that would “incorporate” a wider array and interaction of climate variables?

L183-184: Is there a reference for this idea of “useful precipitation”? Is the range of useful precipitation going to change with potential increasing temperatures (e.g., to include May and September), and if so, is this change included in the modeling exercise?

L199: appears to be an extra word here; check and revise.

L215: Need to include the distance measure and linkage method used in the cluster analysis, and justify those choices.

L224-225: What is the “American soil database”? Is this the NRCS Soil Survey data? Need to clarify.

L305: So I understand that a subjective cut-off is needed, and the authors selected 15%. I have no real issue with that, but I think it would be instructive to examine how sensitive this threshold is. I would suggest the authors use one or two other levels to examine this sensitivity, such as 10% and 20%, or some other values. Without such information it is difficult to assess how serious the potential change may be.

RESULTS: In general when looking at sugar maple and yellow birch, was there no potential increase in potential habitat suitability for these species in the more northerly locales? If not, may be interesting to speculate why in the Discussion. I would assume the soil types are not suitable, whereas the climate might be.

L373: Suggest replacing “like” with “as.”

L392-393: I don’t follow the statement about the significant proportion of global forests. What is? This needs to be rewritten for clarity.

L397: suggest revising to “climatic gradient in Quebec…”

L414: shouldn’t this read productivity instead of production?

L475: I would suggest revising as I do not consider the type of forest harvesting occurring in these areas to be “exploitation.”

L488-512: This needs to be a new paragraph. Lots of arm-waving here that could be re-written to be more objective in my opinion.

L517: I did not double check the Literature Cited for completeness and accuracy.

Reviewer 3 ·

Basic reporting

The manuscript addressed a timely important issue and it was well written. It had sufficient introduction and background to demonstrate how the work fit into the broader field of knowledge. Relevant literature was appropriately referenced. Figures were relevant with high quality.

Experimental design

I have the following concerns, listed in the order of seriousness (the most serious one comes first):
1) Elevation and soil as predictors can compromise the model predictions for the future climates. Elevation is strongly correlated with climate at large or local scales. Soil type can also be related with climate at a large scale. When these two variables are included in the model as predictors, the contributions of climatic variables to the model are likely to be reduced. As a result, although the overall model accuracy will increase by including these two variables, they will not contribute to the predictions for the future as they do not vary in the future. My suggestion is to exclude these two variables from the model.
2) The spatial resolution of 20 x 20 km for model building and predictions appears too coarse. The article addressed the importance of predictions for forest resources management at local scales. Such a resolution does not meet this need. A spatial resolutions at 4 x 4 km or 1 x 1 km is mostly reported in recent literatures for North America.
3) Only three climate variables used for building the model appear over simplified the relationship between climatic niche and climatic variables. The coarse resolution might help the model to achieve a relatively high accuracies, but it might also over simplified the situation.
4) Most publications are using IPCC AR5 GCMs. It would seem odd still using AR4 GCMs. It is understandable that it is not a small job to extract and downscale AR5 GCMs. However, a recently released tool “ClimateNA” can help to get this job done with ease.

Validity of the findings

No Comments

---

## Round 0.2 · Minor Revisions

Both the reviewers are content with the changes you've made and the manner in which you addressed previous suggestions and queries. Thanks for your hard work on the paper and your very thorough rebuttal. There are three minor points I think should be addressed:

1) Please attend to the minor corrections suggested by Reviewer 2
2) I would suggest that it would be helpful to insert a table comparing the RCP emissions scenarios and the older SRES scenarios you used in your study.
3) I would like to see a brief consideration of how changing the threshold value could affect your results. Given you've completed a sensitivity analysis I think this could be reported in the methods to justify your chosen value of 15% in slightly stronger terms.

Please do remember to proof-read your article carefully before you resubmit.

·

Basic reporting

No comments

Experimental design

No comments

Validity of the findings

The findings are probably as good as could have been done 10 years ago, but their validity now depends on how the climate predictions for this area have changed - presumably improved - since then, which was not obviously assessed. The scenarios are not the issue - the older models can use only a fraction of the information the RCPs provide - but the validity of the GCMs used. I don't think this would be hard to assess. Indeed, someone has probably done it and the results could be cited.

Reviewer 2 ·

Basic reporting

No Comments

Experimental design

No Comments

Validity of the findings

No Comments

Additional comments

Overall, I find the manuscript to be improved as the authors have addressed my earlier comments, as well as those of the other reviewers. For the most part I follow the authors logic on the 15% threshold value, which was a concern of mine with the original manuscript. And while I would prefer to see a comparison of how different threshold values change the habitat for all major species and have that included in the manuscript (which would allow the reader to determine the influence of this value), that is problematic as the paper is already quite lengthy. I think this is probably a decent compromise. I also found the Discussion section to be improved and appreciate the revisions the authors have made.

Some minor editorial issues I found in the revised manuscript:

L93: sentence needs to be revised, especially the clause “…affect species turnover at a site and forest productivity.”

L150-151: Added sentence in the revised manuscript seems out of place. Not entirely sure what point the authors are trying to make here with the insertion.

L215: Use of “their” is vague in this context; suggest revising to something like “The ecological and economic importance of these species cannot…”

L341: Spell out number four

L384: Is there a reference for this data? Is it the Web Soil Survey?

---

## Round 0.3 · accepted · Accept

Many thanks for submitting the revised version of your manuscript and for your thorough response to my previous comments.